# Grief Experiences in Family Caregivers of Children with Autism Spectrum Disorder (ASD)

**DOI:** 10.3390/ijerph16234821

**Published:** 2019-11-30

**Authors:** Jorge Bravo-Benítez, María Nieves Pérez-Marfil, Belén Román-Alegre, Francisco Cruz-Quintana

**Affiliations:** 1Mind, Brain and Behaviour Research Centre (CIMCYC), University of Granada, 18071 Granada, Spain; jbravobenitez@correo.ugr.es (J.B.-B.); fcruz@ugr.es (F.C.-Q.); 2Department of Psychology, University of Cádiz, 11001 Cádiz, Spain; belen.roman@uca.es

**Keywords:** ambiguous grief, sorrow, ASD, caregivers

## Abstract

The main objective of this study was to analyse the experience of grief and feelings of loss in family caregivers of children diagnosed with autism spectrum disorder (ASD), as well as the perceived overload from taking on the primary caregiver role. Twenty family caregivers of children with ASD participated. The family members were assessed using an ad-hoc semi-structured interview that addressed the families’ reactions to the diagnosis, implications for daily functioning, and concerns for the immediate and long-term future of their relatives with ASD. The results indicate that family caregivers of children with ASD endure intense and continuous sorrow and grief due to the impact that having and caring for a child with these characteristics has on all aspects of their lives. These data highlight the importance of creating support and intervention programmes and services focused on the feelings and manifestations of ambiguous grief that occur in these family members, in order to improve their well-being and quality of life and reduce caregiver role overload.

## 1. Introduction

An estimated 1 in 160 children has an autism spectrum disorder (ASD). This is the estimated mean, as the observed prevalence varies considerably across studies, with well-controlled studies reporting considerably higher figures [1]. According to epidemiological studies conducted over the past 50 years, the global prevalence of these disorders seems to be increasing. Some possible explanations for this apparent increase have been suggested, including greater awareness, broader diagnostic criteria, better diagnostic tools, and improved communication among professionals [1].

ASDs are a range of mental disorders of the neurodevelopmental type that appear at an early age and are characterised by difficulties in social interaction, communication, and flexibility, among others [2]. People with ASD have complex care needs and require a range of integrated services, including health promotion, personalised care, rehabilitation services, and collaboration with other sectors, such as the educational, occupational, and social sectors [1].

Planning the needs discussed above starts with one of the key moments: informing the family of the diagnosis. When a family receives an ASD diagnosis of one of its members, a very painful assimilation process ensues, which experts compare to a grieving process, in this case, grief over the loss of a “typical child” [3].

Raising a child with ASD is a challenge for parents, who, after coming to terms with the diagnosis, must begin to implement adaptive strategies that favour the development of their children and the family as a whole [4]. This situation is complex and has been associated with an important change in family dynamics [5,6]. Raising a child with ASD often entails an emotional and financial burden for families, and caring for children with severe cases of ASD can be demanding, especially when access to services and support is inadequate [1].

Researchers have emphasised the occurrence of physical and emotional health problems and high levels of caregiver overload in these children’s parents [7,8,9,10,11]. Parental feelings of disbelief, distress, anxiety, or sadness are common at the time of diagnosis and in the following months [12]. Numerous studies show that these parents often experience higher levels of stress, depression, anxiety, and lower quality of life than parents of children with other types of disabilities, such as Down syndrome, or typically developing children [13,14,15,16].

Another consequence of the child-rearing process is the isolation of caregivers, especially the primary caregivers, who are mostly mothers [17]. This isolation is often reinforced because caregivers, trying to prevent stigmatisation, avoid any social interaction [18]. In addition, caregivers often have higher levels of stress and overload due to the responsibility acquired, thus experiencing feelings of guilt, which are increased when facing criticism from those in their immediate environment when there are problems regarding the child’s upbringing [11].

An important part of the emotional response of these families is associated with feelings of loss or grief [19,20]. Accordingly, Ponte et al. [21] reported that parents experience emotional pain due to the loss of their ideal child and the disruption of their expectations, as well as shock and/or helplessness in family members after diagnosis. Several concepts have been proposed to explain the dynamics experienced by these families in these cases. Among these is the concept of ambiguous loss [22,23]. O’Brien [19] relates this concept to several factors, such as uncertain diagnoses and prognoses, variable daily functioning, seemingly normal development of the child during the first years, manifestation of only some symptoms, and the difficulty of these children in recognising and expressing their emotions. Parents may experience grief over the loss of their “expected child” and find it difficult to accept the reality [24,25]. Parents’ grieving reactions to an ASD diagnosis have also been considered to be a type of “chronic sorrow” [26] and “non-finite grief” [27].

Other authors have adapted the concept of “ambiguous loss” in order to include a fundamental characteristic of this type of circumstance, in which the child, although physically present, appears to be emotionally absent or distant [23]. In this context, Olshansky [26] defines the concept of chronic sorrow as a “profound sadness,” referring to the sadness felt by the parents of children with mental disabilities [28]. Boss [23], in turn, describes ambiguous loss as an “incomplete and uncertain loss” which occurs in two different ways: the family member may be physically present and psychologically absent, or physically absent and psychologically present. Finally, Bruce and Schultz [29] propose the concept of unresolved or non-finite grief to describe families experiencing loss due to chronic illness, disability, or accident. These authors found that families did not experience the stages of bereavement only once; rather, they continually revisited the different stages again and again, especially when faced with developmental milestones that were not being reached by their children [27].

There are few recent studies exploring grieving reactions of parents of children with ASD [28,30,31,32] that have used grounded theory to analyse parents’ feelings of sorrow and loss, a relevant issue given the considerable emotional adjustment that these parents have to undergo [6]. These feelings could be related to receiving the diagnosis, hopes for the child’s development, how they perceive positive aspects of the child’s education [33], and different coping styles [34,35]. Interventions to support these parents require knowledge of their emotional state in order to plan and implement programmes adapted to their real needs.

The main objective of this study is to analyse the experience of grief and feelings of loss in family members of children diagnosed with ASD, as well as the perceived overload of taking on the primary caregiver role. Identifying this type of loss, as well as the associated emotions and barriers to processing it, is an essential step towards preventing this grief from becoming chronic. This information, together with the information on caregiver overload, could be of great utility in designing programmes to address the specific needs of this population. The specific objective was to analyse the following: the manifestations of grief in the relatives of individuals with ASD; their reactions to the diagnosis (associated thoughts and emotions); and how this has affected their lives (family, work, social, as well as relationships and personal changes).

## 2. Materials and Methods 

### 2.1. Participants

The study involved 20 family members (4 men and 16 women) who are the primary caregivers of individuals with ASD. These individuals were either users of two mental health clinics in the provinces of Cádiz and Granada, Spain, or members of an association of relatives of individuals with ASD in Cádiz. The median age of these individuals was 40.00 years (range: 31–68). Two of the caregivers had completed only basic education, twelve had secondary education, and six had higher education. Fifteen of the participants were married, three were separated/divorced, one was single, and one was widowed. Regarding their employment status, 10 were employed, eight were unemployed, and two were engaged in household chores. The age range of the children at the time of the study was from 6 to 12 years of age (mode = 8). All of the children were attending school and lived with their original families (15 with both of their parents and 5 with their mothers) (see Table 1).

In the present study, the children’s median age at which the parents were informed of the diagnosis of ASD was 30.00 months (range: 17–132). The professional who communicated the diagnosis was a psychologist on 10occasions.Medical professionals (psychiatrist, neuropaediatrician, paediatrician, neurologist, and general practitioner) communicated the diagnosis on eight occasions. The diagnosishad been communicated by a counsellor in one case only. Only one individual in the sample reported having discovered the diagnosis themselves, without being informed by third parties (see Table 2).

### 2.2. Instruments

Family members were assessed using an ad-hoc semi-structured interview that addressed family reactions to the diagnosis, implications for daily functioning, and concerns for the immediate and long-term future of their family members with ASD.

Subsequently, family members were interviewed. All of the interviews were conducted by the same trained researcher, specialized in the field of health psychology, on the mental health clinics and associations’ premises, in a quiet environment ensuring privacy. The interviewer is an expert in qualitative methodology. In addition, a pilot study was conducted, which consisted of interviewing two professionals from the participating centres. The one-to-one interviews were semi-structured and largely based on previous research in the area. Although the main aim of the interview was to explore patterns in going through the grieving process, it seemed sensitive to gather further data on other issues, such as information about the moment of receiving the diagnosis, the caregiver’s overload or the change in future expectations. Table 3 shows the general framework of the four sections in which the interview was structured.

### 2.3. Procedure

Firstly, the management teams of the different centres were contacted in order to present them with the research objectives and ask for their collaboration. Once approved, the study was disseminated by means of a dedicated website and leaflets, as well as by the participating centres. Centre staffs were informed of the research objectives and they subsequently contacted the participants. The participants were called to a meeting where the principal investigator informed them about the study and answered their questions. Interested individuals signed the informed consent form. They were also provided with a copy of the protocol for handling of personal data. In the information sheet, participants were provided with general data of the research, the voluntary nature of the participation, the commitment acquired when participating, possible risks and benefits for the participant, the confidential nature of the data, the rights of the participant, and the data of contact of the main researcher. Subsequently, family members were interviewed. All of the interviews were conducted on the mental health clinics’ and associations’ premises, in a quiet environment ensuring privacy. The duration of the interviews was variable, the mean being around 30 minutes. The procedure included providing emotional support after the interview to the family members who needed it or requested it. However, in no case was it necessary to do so.

Each of the family members was assigned a numerical code. To preserve their anonymity, their names were not transcribed. The study had been approved by the Ethics Committee of the University of Granada (Ref: 359/CEIH/2017).

### 2.4. Data Analysis

A qualitative design was used based on the grounded theory of Glaser and Strauss [36].

The analysis of the interview data was carried out with Atlas.ti v.7 (Scientific Software Development GmbH, Berlin, Germany). The interviews were recorded on paper, taped, and then digitally recorded and transcribed verbatim using f4 software. The main ideas present in these data were then underlined, coded, and grouped into categories. Finally, the hierarchical relationships of the codes and families were created, building a network map.

Specifically, a method of constant comparison was used, organized into three stages, adapting the proposal of Strauss, Corbin and Zimmerman [37] to the present research objectives. First, an inductive strategy was used in which fragments of the participants’ discourse were analysed and codified (micro-analysis). The properties of each code were later described and delimited, allowing for comparison among them. Names for these codes were selected based on expressions in the participants’ discourse. This step represented the open coding stage, in which four researchers participated. The different codes were named and defined by consensus. In cases of disagreement, debate among the researchers and a review of the literature were conducted to ensure that each code was as close as possible to the experience of the participants. Second, the identified codes (see Table 4) were grouped into categories according to their characteristics. These categories were derived from the participants’ discourses and were created by comparing cases and merging codes. The most complex categories were organized into subordinate concepts or sub-categories (axial coding). The same strategy as in the previous analytic phase was adopted for grouping the codes. Finally, the categories obtained were integrated, and hierarchical relations were discussed. A central (core) category was identified and the most important codes were used to build a theoretical framework to explain the phenomenon. A core category must be related to other categories, must appear frequently in the data, and must provide a coherent explanation of the data [37]. Later, the theory was refined using strategies of constant comparison. The final model was submitted to triangulation among all of the investigators to ensure its suitability for the phenomenon under study. This final stage corresponds to selective coding as described by [37]. The interviews were transcribed by the researchers with wide experience in qualitative investigation. The analysis was performed using the original Spanish interviews. The quotations selected for inclusion in this article were independently translated by two experienced professional translators (English-native and Spanish-native), who then produced a jointly-agreed version that was further reviewed by two of the authors (both fluent in English) to ensure that the meaning of the original was accurately conveyed.

## 3. Results

Table 4 shows the identified themes associated with the grieving experience of family caregivers of people diagnosed with ASD: anticipation, the children’s age, professional who communicated the diagnosis, reaction to the diagnosis, disruption of expectations, experience of grief, discontinued activities and time for oneself, changes in family dynamics, and feelings of loss/gain. These themes were coded and grouped into three categories: diagnostic processes, grief processes, and stress and overload.

### 3.1. Codes Grouped under The “Diagnostic Process” Category

#### 3.1.1. Anticipation

A child’s acquisition of different psychomotor, cognitive, social, and linguistic skills is something the family looks forward to during the child’s first years of life. Families usually know the normotypical age at which the different developmental milestones are reached, sometimes comparing the child’s development with that of another child of a similar age, or with a older sibling.

Most of the mothers had already been observing certain signs that made them suspect and/or become alerted to the possibility that the child might be affected by some disorder. Consequently, they had been expecting it by the time the diagnosis was communicated to them. In these cases, this anticipation cushioned the impact of the diagnosis.

“I already knew, I saw things in my son, so I expected it.” (06-Mother) 

“At 18 months, but I knew since he was born.” (08-Mother) 

“B. His wife had already noticed since he was a very young boy. She knew that something was wrong with him […]. It was a major blow, but I already imagined it because I saw it. Because he had been doing very weird things since he was a little boy” (12-Father)

“It was kind of what we had been expecting. It was not a slap in the face because we had suspected it” (15-Father).

#### 3.1.2. Age and Professional Who Communicated the Diagnosis

ASD is usually detected before the age of 3. It is therefore common for family members to begin searching for explanations of the warning signs they see in their child before this age. This search usually involves seeking advice from different child health professionals (psychologists and paediatricians, mainly) on the child’s development, and concludes when the diagnosis of ASD is confirmed by the professional.

### 3.2. Codes Grouped under The “Grief Process” Category

#### 3.2.1. Reaction to The Diagnosis

Receiving such a diagnosis is usually not easy, and it goes without saying that this is always coupled with an emotional response. The different reactions to a diagnosis can be shaped by various factors, such as the coping skills of family members, predictability, prognosis, severity, and availability and knowledge of social and financial resources.

The families of the children with ASD expressed having experienced intense emotions at the time of diagnosis. The reactions they described of that moment were diverse, ranging from a state of shock to a proactive and resilient attitude, through phases of denial, guilt, anger, and helplessness (see Table 5).

#### 3.2.2. Disruption of Expectations: Differences Regarding the Children’s Future.

There was great homogeneity with respect to the disruption of expectations, specifically with respect to how family members expected their child’s future to be before the diagnosis and how they believe it will be after the diagnosis. At the time of pregnancy or during the first year of life, the families of children with ASD did not imagine that their child would receive this diagnosis. As a result, they imagined a normally developing child with an ideal future during this time. These expectations disappear the moment they become aware of the full implications of the diagnosis. We observed different emotional reactions in parents to this mismatch between the ideal and the real. Concern and uncertainty were the most frequent emotions among the sample of family members. Both emotions seem to be related to the ambiguity about the prognosis and the future of the child. In particular, the main concern of relatives was the independence and autonomy of the child with ASD (see Table 6).

#### 3.2.3. Grieving Experience

The experiences of diagnosis-related grief in the families of children with ASD who participated in this study are varied. Most family members view the process of assimilating their child’s diagnosis as an experience as intense as grieving over the death of a family member, or even more painful, because of the lack of predictability and the uncertainty about the future. Other family members, on the other hand, highlight that the difference between grieving for the death of a loved one and grieving for a diagnosis of ASD lies in the cyclical nature of the latter, which is characterised by a fluctuation of painful emotions, such as anguish and sadness, and periods of acceptance and happiness. Other family members report that this feeling of loss is caused by the difference between the expected child and the real child. Although after receiving the diagnosis the child with ASD does not actually die, the “healthy” child with whom they had been living until that moment is gone. For others, grief is related to a chronic sorrow, an “endless, living sorrow.” Finally, other family members experience this as an ambiguous loss, because they feel as if the child has been taken away from them and that the child may never speak to them (see Table 7).

### 3.3. Codes Grouped under the “Grieving Process” Category

#### 3.3.1. Discontinued Activities and Time for Oneself

The alterations in different aspects of a child with ASD require multidisciplinary treatment involving different types of professionals, such as psychologists, educational psychologists, speech therapists, and occupational therapists, as well as school hours at the child’s own chronological age.

Family members of children with ASD are engrossed by the disorder and the intense pace of life involved in raising such children. This means that they have to give up many of the activities they used to engage in.

“I was leading a completely different life: travelling, leisure...” (02-Mother)

“Going out to eat with the family, going to funfairs, going for coffee... going out and leisure in general. Activities for couples too.” (04-Mother)

“Many: travelling, going out to bars, the gym, sports...” (11-Father)

“Everything: working, having fun, going for coffee, going out, having a drink, enjoying bars...” (18-Mother)

“Sport, I loved it... and now I don’t do anything. Cinema, reading... everything.” (19-Mother)

Along the same lines, when family members were asked about the time, they spent on themselves during the week, the responses were very similar and showed an evident lack of time spent on themselves.

“Little. The time to shower. And when they’re at school I have chores and things to do.” (01-Mother)

“No, not at all. I want to make time to walk because I need it.” (03-Mother)

“Two hours a week in the gym. Also going shopping alone is an escape route for me.” (05-Mother)

“When I’m sleeping. Other than that, not at all. That’s my hobby. My coffee and my cigarette are the short moment I have for myself.” (08-Mother)

“When his grandparents take him with them, once a month or so. That day that I have off I’m doing garden stuff or crafts. And that’s the only time I can dedicate to myself.” (12-Father)

“Nothing, I don’t spend time on myself. I have long hair because I can’t go to the hairdresser.” (16-Mother)

#### 3.3.2. Changes in Family Dynamics

Children with ASD have a number of needs that differentiate them from normotypical children. They will have to attend certain activities and will require certain support measures and therapies that are not necessary for children without ASD. The deficit in communicative and social functioning that characterises children with ASD affects social interactions and differentiates them from children without ASD, thus compounding the situation. This is why family members of children with ASD, when asked about changes in family dynamics, agree that, when there is an individual with ASD in the family, everything revolves around the needs of this individual, thereby affecting family dynamics completely.

“It limits you. A typical mother would go to the park in the evenings, we don’t.” (01-Mother)

“Totally. We do everything through A. If he’s happy, we’re happy. Our lives revolve around him.” (02-Mother)

“Our outings are adapted to him. For example, my daughter loves McDonald’s, and instead of eating in, we order it at the drive-thru.” (04-Mother)

“Everything is crazy, from the time you get up to the time you go to bed. There is nothing normal about meals, outings, tasks, conversations...” (07-Mother)

“It has 100% affected everything. Now everything revolves around him: now he’s got this therapy, these activities...” (11-Father)

“All the time devoted to him. I also have a girl. I used to take them to classes and stuff... and she couldn’t take it anymore. She needs time for her studies and it’s a pain. I have a kid coming home to help.” (13-Mother)

#### 3.3.3. Differences, According to Sex, as to Whether There Is Anything Family Members Would Like to Recover

Only in the sample of women have we found feelings of loss of freedom, time, etc., as a result of the diagnosis of their children. The women have reported wanting to recover certain things in their lives, while the majority of the sample of men answered that there was nothing they would like to recover (three fathers out of four). In contrast, most women report lack of time and overload, as well as anxiety and constant worry.

“The harmony when they were babies, not thinking.” (04-Mother)

“My freedom.” (05-Mother)

“More time for myself.” (06-Mother)

“The life I had before; with my books, my gym, friends... having my own small space.” (09-Mother)

“Knowing that my son doesn’t have anything and not having to worry so much about him.” (10-Mother)

“The energy of youth.” (20-Father)

#### 3.3.4. Gains

When a traumatic event occurs in the family, sorrow is inevitable at first. Over time, family members usually undergo a process of acceptance of the event and develop a resilient attitude that leads them to value the positive side of the situation. However, this process is influenced by different factors related both to the event itself and to the characteristics of family dynamics.

All members of the sample in this study acknowledge that, despite having lost many things, such as time, leisure, or peace at home, they have gained a child, strength, and family unity.

“Almost everything. But I have gained my son.” (02-Mother)

“But I’ve gained finding myself. I have gained the ability to face it and say that my son has ASD in front of people. I used to be very embarrassed.” (04-Mother)

“The change has been for the better. It has brought us closer together. Now the bond with her (my wife) is stronger.” (11-Father)

“Not a thing. In fact, I feel that I have gained from this experience.” (12-Father)

“A little bit of freedom, but I’m delighted to be with my boy.” (15-Father)

#### 3.3.5. Integration of Results

The results show that there is a time sequence that begins with the anticipation of the diagnosis, before receiving it officially. Once the diagnosis has been communicated, family members react in a variety of ways. Among these reactions we identified the phases of shock, denial, guilt, anger, fear, or a positive attitude. The experience of grief is varied and ambiguous, including the disruption of expectations in virtually all cases, as well as the existence of various feelings, such as uncertainty, worry about the future, guilt, and, in some cases, optimism.

After diagnosis, the lives of these parents are completely changed. They become overburdened and exhausted, with little time for themselves, having to give up many of the activities they used to do, and suffering significant changes to their family dynamics, which from now on are completely focused on the needs of the child with ASD. This stress is compounded by being in charge of more family members, i.e., other children who are normally developing, but who also need their attention, or other family members who need their attention. This means that many areas of the lives of these family members are affected, such as leisure and time for themselves, their relationships with other family members, or family dynamics itself (see Figure 1).

## 4. Discussion

The main objective of this study was to identify the manifestations related to the grieving experiences of primary caregivers of children with ASD and their perceived overburden from adopting the caregiver role. The diagnosis of ASD of a family member is associated with important changes in family functioning, as well as with the need to face new challenges in caring for that child, which puts significant pressure on caregivers, with ramifications for their physical and mental health.

Participants in this study acknowledged experiencing diverse and intense emotions related to grieving for having a family member with ASD and to having high levels of overburdening for caring for and raising these children, which is consistent with previous studies [13,16,30,32]. These emotional reactions include anxiety because of uncertainty, sadness and guilt because of the loss, or anger because of difficulty accepting the situation. These emotions can promote the onset of physical and mental health problems [7,8,9,10]. Different types of grief have been identified from the qualitative analysis of the interviews with family members. Unresolved grief, cyclical grief, and ambiguous loss stand out. This illustrates the complexity of these experiences and the need to take into consideration the specific characteristics of each experience.

Traditional models of grief are difficult to apply to people experiencing this type of loss [38]. The particular characteristics of these feelings of loss and grief, as well as their continuation and perpetuation over time [6], make it necessary to implement specific intervention programmes that address these feelings and emotions in order to improve their well-being and quality of life. These programmes would focus on the following: helping caregivers to accept the reality of the situation; tools to identify their feelings; emotions and sorrow caused by the loss, in order to work on them; strategies to adapt to the new unexpected situation; and identifying new goals and objectives beyond the caregiver role [39].

Early diagnosis is extremely important in order to minimise its impact and emotional effect on parents, as well as to establish criteria for early intervention in children [40].

When the diagnosis is communicated to them and they become aware of all that it involves, different emotional reactions are produced, such as worry, uncertainty, disappointment, and resignation. Voguel [41] points out that mothers, after the diagnosis, are forced to bid farewell to the mother–child bond that they had envisaged, and when mothers feel that the child they had imagined is gone, they experience a psychological death. Our results indicate that the main aspect that characterises and differentiates this type of loss from others is that the child has not died, although the family’s dreams, expectations, and hopes for the child have died [29,38,42]. In this sense, the subjective experience of feelings of loss and grief in parents with children with ASD is manifested, to a large extent, over the loss of a child that was normal and that, at a time of its early development, was lost, but not over the expected ideal child and the dreams and hopes placed in this child [24,25,33]. However, it should also be noted that some parents display an optimistic attitude after learning of the diagnosis. In this sense, they express that knowing their children’s real situation and needs allowed them to start an active coping mechanism and seek solutions suited to the new reality.

With regard to the experience of grief at the time of diagnosis, family members reported that they felt an experience as intense as that of grieving for the death of a family member, or even more painful, alluding to the uncertainty and lack of predictability of the future. Other relatives stated that they felt their grief was different from that of other types of losses, because of the cyclical nature of their grief or the disruption of their expectations about the future. This diagnosis would entail for families the loss of a previously “healthy” child, only to be mired in uncertainty about their child’s prognosis and development [41]. These results are in line with the concepts of “chronic grief” [26], “unresolved grief,” [27] and Bruce’s and Schultz’s “non-finite” grief [29].

Relatives report in their discourses that they are caught up in and trapped by the disorder and by the hectic pace of life involved in raising a child of this nature. As a result, they have been forced to give up many of the activities they used to do, the friends they once had, and the lifestyles they once enjoyed. They also commented that they do not have time for themselves, that they have neglected themselves, and that they prioritise the needs of the child over their own. All this is a result of the complexity of raising a child with ASD and the profound impact and change it produces in family dynamics [5,6]. The characteristics and particularities of these children make them have a set of needs that have to be met and that are very demanding of others, such as attending certain stimulation activities (playroom, swimming pool, horseback riding, surfing, etc.) and specific therapies (with the psychologist, speech therapist, or occupational therapist, among others) [7,8,9,11,13,16].

In spite of all this, some family members show signs of resilience and recognise they have also gained from this experience. Resilience becomes a fundamental element for coping with the problems these families have [43]. It is very helpful to encourage family members to listen, express their opinions and needs freely, seek support from their community and/or loved ones, and use constructive problem-solving strategies. These findings are consistent with Walsh’s theory of family resilience [44]. This theory states that individual resilience is connected to the resilience of the whole family, which is the positive adaptation to a significant adversity [44,45,46,47].

The implementation of specific intervention programmes focused on the emotional and coping difficulties experienced by these family members is crucial, since the emotions and sorrow involved in this type of grief are compounded by the great complexity involved in raising children with ASD [5,6]. Family members encounter many obstacles that interfere with an acceptable quality of life [48], from the initial diagnosis to the caregiver’s ongoing responsibilities and daily demands. The consequence of not addressing these issues is that the tensions from caring for a child of this nature will increase stress levels in parents, which will lead them to use maladaptive strategies to cope with these tensions. Seymour et al. [49] point out that the consequences of using these maladaptive strategies will be manifested in the behaviour of family members, by neglecting their responsibilities at work and at home, or by opting for problem-solving strategies that are unfavourable or cause more emotional problems. All of this will result in the occurrence of negative and aversive feelings towards the child with ASD [49]. In general, we have found that, in order to face the stress caused by these changes, family members use maladaptive strategies, such as avoidance, self-blame, drug abuse, and trying to deny the reality by recreating in their minds these stressful situations in a more positive but unreal light [50,51].

The grief felt by family members of people with ASD is called disallowed grief, and it has a number of characteristics that differentiate it from other types of grief. This type of grief is dismissed by the environment or society surrounding the bereaved. Manifesting this type of grief is denied to them, not valued, and even reproached to and blamed upon the bereaved, which prevents them from receiving social support from their environment. This is compounded by the difficulties inherent in individuals with these characteristics, which make it impossible for their relatives to complete their grieving process, because grieving never ends. This is due to the fact that successive losses are experienced throughout the development of the child, so that in each milestone not reached, there appear feelings of loss, frustration, pain, injustice, uncertainty, anger, and sadness 

In summary, the following may be highlighted among the main contributions deriving from this study: a) the importance of making the diagnosis as soon as possible in order to allow family members to accept the disorder and cope with the new situation as soon as possible; b) the importance for these families of being able to identify some of the manifestations they experience as part of their grieving process, which is fundamental to overcoming it; c) the importance for professionals to be trained and proficient in identifying and recognising the emotional impact generated by communicating an ASD diagnosis, and to be able to assess associated emotions and, if possible, to monitor these emotions; and d) the requirement to address the needs that have been identified in the parents’ discourses. These needs are related both to the normalisation of the grieving process (which in most cases is not recognised as such) and to the provision of coping strategies regarding the management of the disorder and the levels of caregiver overload.

The main limitation of this study is the reduced number of participants in the sample. It would be necessary to increase the sample in future studies, especially that of fathers. It would also be interesting to be able to compare the discourse of relatives who are members of associations with that of other caregivers in order to verify to what extent these associations help to alleviate the emotional discomfort of these caregivers. It would also be desirable to involve siblings and other immediate family members. A further major limitation of the present work is its cross-sectional nature. It is necessary to continue to make progress in identifying the specific characteristics of this type of grief by focusing on its cyclical nature. To this end, it would be important to carry out longitudinal studies with long-term monitoring. In addition, the sampling method used was not randomized. Rather, intentional sampling was used, thus limiting the application of the results to other settings. Another limitation of this study is the low number of male participants. Finally, differences in the severity of the diagnoses could not be taken into account in order to assess their effect on the grieving process. As a result, caution must be exercised when interpreting the results obtained and generalising them to other populations in different cultural settings.

## 5. Conclusions

In conclusion, it is worth pointing out the profound impact of raising a child with ASD on physical and mental health of caregivers, as well as on financial and practical aspects of their lives. The characteristics of the symptoms, together with the uncertainty and lack of predictability of the course of the disorder, mean that feelings of sorrow and grief do not cease and are repeated and continued at each unattained developmental stage of the child. All of this underlines the importance of creating programmes and services to support and address the feelings and manifestations of ambiguous grief that are produced in these family members in order to improve their well-being and quality of life, and to reduce the overload in the performance of their caregiver role.

## Figures and Tables

**Figure 1 ijerph-16-04821-f001:**
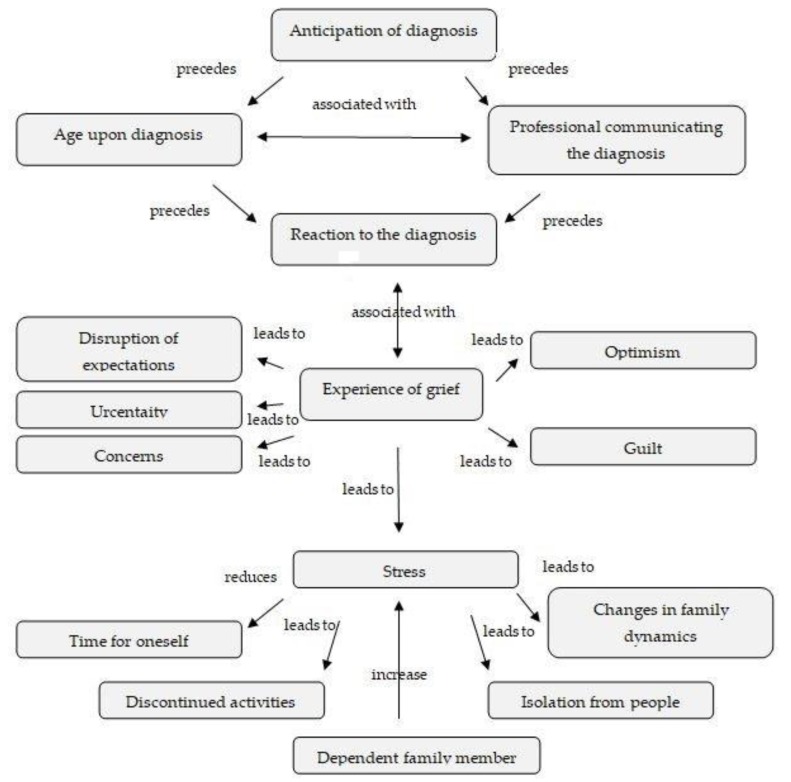
Concept map of the emotional impact of the diagnosis on parents of children with ASD (autism spectrum disorder).

**Table 1 ijerph-16-04821-t001:** Socio-demographic data of the participants.

Subject	Sex	Age	Level of Education	Marital Status	Employment Status
01	Female	36	Secondary education or vocational training	Married	Unemployed
02	Female	32	Secondary education or vocational training	Married	Unemployed
03	Female	40	Secondary education or vocational training	Separated/divorced	Unemployed
04	Female	40	Secondary education or vocational training	Married	Employed
05	Female	57	Secondary education or vocational training	Widow	Employed
06	Female	45	Graduate	Separated/divorced	Unemployed
07	Female	42	University studies	Married	Employed
08	Female	31	Secondary education or vocational training	Married	Unemployed
09	Female	40	Secondary education or vocational training	Married	Unemployed
10	Female	48	Secondary education or vocational training	Married	Employed
11	Male	37	Graduate	Married	Employed
12	Male	35	Secondary education or vocational training	Married	Employed
13	Female	45	Basic education	Married	Unemployed
14	Female	30	Secondary education or vocational training	Single	Employed
15	Male	40	Basic education	Married	Unemployed
16	Female	40	University studies	Separated/divorced	Employed
17	Female	68	Secondary education or vocational training	Married	Household chores
18	Female	40	Secondary education or vocational training	Married	Household chores
19	Female	43	Graduate	Married	Employed
20	Male	45	University studies	Married	Employed

**Table 2 ijerph-16-04821-t002:** Children’s ages, in months, at which the diagnosis was communicated to their family members and the professionals who communicated it.

Subject	Age (months)	Professional
01	36	Psychologist
02	24	Psychiatrist
03	27	Psychologist
04	18	Herself
05	54	Psychologist
06	23	Psychologist
07	17	Neuropaediatrician
08	18	Psychologist
09	132	Counsellor
10	120	Psychologist
11	18	Paediatrician
12	24	Psychologist
13	42	Neurologist
14	36	General practitioner
15	24	Psychologist
16	30	Psychiatrist
17	-	Neurologist
18	84	Psychiatrist
19	60	Psychologist
20	66	Psychologist

**Table 3 ijerph-16-04821-t003:** Sections of the semi-structured interview.

**Section 1:** Diagnosis	Who communicated the diagnosis to the relative? At what age? How did you feel at the moment you were communicated? What were your reactions? What concerns came to mind?
**Section 2:** Loss and grieving process	Interference in work, leisure, friendship, family, care people and relationship, health, welfare in general. Associated emotions. Perceived losses.
**Section 3:** Caregiver overload	Activities that you used to do and have stopped doing now. Do you spend time for yourself? How much per week and how? How has it affected family dynamics?
**Section 4:** Future	Has your vision changed about the future of your family member due to diagnosis? Could you explain it?

**Table 4 ijerph-16-04821-t004:** Themes identified and associated with the grieving experience of family caregivers after receiving the diagnosis of autism spectrum disorder (ASD).

DiagnosticProcess	Grief Process	Stress and Overload
Anticipation	Reaction to the diagnosis	Discontinued activities and time for oneself
Age of the child	Disruption of expectations	Changes in family dynamics
Professional who communicated the diagnosis	Experience of grief	Feelings of loss/gain

**Table 5 ijerph-16-04821-t005:** Quotations associated with reactions to the diagnosis.

Subcode	Quotations from the Participants
Shock	“In shock. The stereotype of autism that I had was more exaggerated, so I didn’t think my son had this.” (07-Mother) “I didn’t know how to react, I was shocked. I didn’t know what to do, or how to act, or... this is horrible... unbearable...” (17-Grandmother)
Denial	“Petrified. I thought: This is not happening to me. And then I thought: Why me? What will become of my son?” (05-Mother)
Guilt	“My world fell apart.It crossed my mind whether I had been a good mother.” (09-Mother)
Anger	“I felt an intense rage and then frustration afterwards: Why me? I felt helpless not knowing how to deal with the situation... uncertainty.” (11-Father) “Awful, I don’t want to say it because I’m going to cry. I felt rage... why did this happen to me?!” (18-Mother)
Proactive attitude	“I felt like I wanted to fight. I thought: now that I know what my child’s got, we’re going to work on this. I reacted proactively. I got down to work: looking for information and resources everywhere. I avoid standing still.” (03-Mother) “I felt that you have to try harder, press harder, and take more care of the situation.” (20-Father)

**Table 6 ijerph-16-04821-t006:** Main quotations on the expectations for the future of the child in the past and in the present.

Emotion	Quotations
Concern and uncertainty	“A healthy girl who would run and jump, happily. Now I don’t know, I’m not quite sure. Her autonomy is what worries me the most.” (01-Mother) “Now his future is a question mark. It depends on his progress. What I want is for him to be independent and to be able to lead a life like everybody else.” (02-Mother) “Now I’m very worried about my son’s future. That’s what worries me the most, if he will be able to become independent, autonomous. I’m worried that society will take advantage of him […]. I’m very concerned that there will be no more resources for him, that there are finite resources.” (07-Mother) “A normal future, with a job... Now I really don’t know. That’s what worries me the most.” (09-Mother) “Very different. I imagined that I was going to be able to talk to him, that I was going to teach him my stuff, sports... and that he was going to have an education. Now, I really don’t know what’s going to happen, if he can’t speak...” (12-Mother) “I didn’t used to think much about the future, but I imagined him beingwell, happy. Now I don’t know, I’m hopeless. Sometimes I think about giving him a sibling, but I’m afraid.” (14-Mother) “The usual... where is he going to study? Will I teach him English?... The usual... Now, I’m scared and you think what will become of him when I’m gone...? The million dollar question.” (16-Mother) “With my wife, several children, a simple life, and making a living. Now it hurts because I wanted to have another child and take more care of them. And I’m concerned for the day we are gone.” (20-Father)
Disappointment	“We all expect to have a perfect child, and that’s the letdown we get. Because that’s the child we lose.” (04-Mother) “I thought that he would be a university student, that he would play football, be a good person, an athlete... now I think of him as totally dependent, always asking to have good people around to take care of him.” (05-Mother) “Before, when he was younger, it came to mind a lot what it would be like without the diagnosis. I really wanted to have a child to enjoy it: when they started to walk, to talk... and I’ve been left with the desire to enjoy my son’s normal early stages.”(07-Mother) “I thought that he’d be an amusing kid, very talkative… a little rascal” (08-Mother) “I wanted some intellectually gifted children, super clever, and now what... My friends, with their children, who are super good at something whereas mine, what... well...” (13-Mother) “[…] that now you have to be dedicated to him for everything, I imagined him on a motorbike and now... you see... not anymore.” (15-Father) “Like everybody else, I wanted to have three children, very nice, like everybody else... my studios arein Madrid... And now I’m gloomy, always with my son, he is totally dependent and I don’t trust anyone.” (18-Mother)
Resignation	“I pictured him studying, with his friends, with his girlfriend... a normal life. Now, I don’t even think about the future, what for?” (06-Mother) “Well, having my grandson, pampering him, taking him to the streets, to the cinema, drawings... talking to him... now I don’t have a future, I don’t want to think about it, it scares me so much...” (17-Grandmother)
Optimism	“I wanted my children to travel, to be independent, to go to university... to be free people. Now I think the same way, or at least I hope so. It’s improving a lot.” (04-Mother) “Like anyone else: university studies, football... now I still think the same way, or I’d like to think the same way. Instead of football, chess... let him be happy with whatever it is. If instead of football he likes to learn the roads, so be it.” (11-Father)

**Table 7 ijerph-16-04821-t007:** Main quotations about feelings and experiences related to grief.

Emotion	Quotations
Unresolved grief	“For me the grief is worse now than at the time of diagnosis, because I see how L. is and I’m worried about the future.” (01-Mother) “It’s been worse for me. Because when I lost my parents, it hurt, but it’s something I had accepted would happen. But not this.” (05-Mother) “Yes, to me it’s something similar. I keep experiencing it inside. I experienced it in this way: they give you the diagnosis and you have to ride it out. I made sure I didn’t have time to grieve by looking for therapies and resources everywhere.” (11-Father)
Cyclical grief	“It’s not the same to me. Grieving for a diagnosis of ASD is a cycle of grief that you enter and exit continuously. Each stage of the child’s development makes you worry about some things and it makes you grieve again.” (07-Mother) “Yes, it’s similar. It changes everything for you. It’s as though I had to learn everything from him. I have to learn to deal with him. And then there are days when you just can’t accept that your child has ASD. You have ups and downs. The grief goes up and down, up and down.” (08-Mother)
Grieving for the loss of the expected child	“Yes, it’s not the same because the person is there, but it’s not what you had in mind. When you consider having a child, you don’t expect this. And I really thought about it.” (01-Mother) “Yes, it’s as if you’ve lost your child. We all expect to have a perfect child, and that’s the letdown we get. Because that’s the child we lose” (04-Mother) “It may resemble something like that, because what you imagine when you’re having a baby vanishes, so it’s like you’re losing something of yourself.” (12-Mother) “It’s not like a bereavement, no. It’s like... now what? Uncertainty. I wanted to have intellectually gifted children, super clever, and now what? …” (13-Mother)
Chronic sorrow orlatent grief	“It’s very similar, but it’s more painful. Because when you lose someone you learn to live without them, and now you learn to live with them, it’s exactly the opposite.” (06-Mother) “No, not so much, but I’m always trying to make sure that he does not suffer, that people don’t deceive him... it hurts so much that I have inappropriate outbursts at times.” (10-Mother) “No, it felt worse than the death of a loved one. My father died, I have already experienced that, but this is worse than the death of a loved one. It’s like a living sorrow. It never ends.” (16-Mother) “No, I feel it more (worse)... because this is not sorrow, it’s... just that it leaves you empty, a very big void... there is sorrow here, there is no satisfaction... a very big sorrow in your soul.” (17-Mother)
Ambiguous loss	“I didn’t experience it that way. Although it is as though he had been taken away from me. After the vaccinations, he stopped looking at me, talking to me... everything. Before, he used to do it.” (03-Mother) “No, the feeling is that your son is never going to talk to you. It’s a weird feeling, I haven’t experienced anything similar, but it’s not comparable to the loss of a loved one.” (15-Father)

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
