# Peer review of "Grief Experiences in Family Caregivers of Children with Autism Spectrum Disorder (ASD)"

_ijerph, 2019, doi:10.3390/ijerph16234821_

Round 1

Reviewer 1 Report

This is a well written account of a study undertaken with Spanish  parents who have a child diagnosed with autism.  I welcome studies undertaken in non English speaking countries as it helps to identify the universal impact of autism on parents and families.  That said the present study has a number of weaknesses that the authors may be able to address.

They have not identified the specific aims of their study and in particular how it might add to our understanding of a grief response in parents.

Linked to this point, more details are required on the questions and prompts  asked of parents during the interviews.  Did the same person conduct all the interviews and what prior contact did they have with the parents?

What information was given to parents about the reasons for undertaking the study and how their information would be used?  What additional support was available to parents if they found the interview process upsetting?  Did any parents become emotionally upset?

The sample of parents span a wide range of ages but no information is to given about the ages of children nor of their current circumstances, for example are they living with their parents?   Although information is provided about the professional who gave the diagnosis, it is more crucial to know how many years ago the diagnosis was given. Although I appreciate that grief reactions can re-occur, I hope there were a majority of parents for whom diagnosis had occurred in the recent past.  Was this a requirement in the recruitment strategy?

More detail is needed about the approach used in the qualitative data analysis, such as how grounded theory was undertaken?  What evidence did the authors find that grief emerged  as a core theme in parental responses?  Were there instances of   how parents coped with their grief?   Also what checks were used to assess agreements across raters and were the emerging themes confirmed with parents?

The discussion is largely a recap of the findings when it should be more focused on the specific aims of the study (see previous comment) and the implications for services and professional involved in conveying a diagnosis to parents in both the immediate term and also over time.  However they also need to discuss the value of utilising grief as an explanatory concept for parental responses when there are other frameworks available, especially ones that relate to present circumstances rather than events that may have taken place many years previously

Under limitations, I suspect that if some of the comments noted above cannot be addressed, they should be added to this section.

In the conclusion section, the authors might provide more detail on what specific actions services need to undertake in support of families.  In many countries, their focus is largely on the person with autism.

Author Response

We are very grateful to the reviewers for their careful reading of our manuscript Ambiguous grief experiences in family caregivers of children with autism spectrum disorder (ASD).

We include below our point-by-point responses to the reviewers’ comments with an account of the corresponding modifications.

Thank you for your attention.

This is a well written account of a study undertaken with Spanish parents who have a child diagnosed with autism.  I welcome studies undertaken in non English speaking countries as it helps to identify the universal impact of autism on parents and families.  That said the present study has a number of weaknesses that the authors may be able to address.

They have not identified the specific aims of their study and in particular how it might add to our understanding of a grief response in parents.

Indeed, no specific aims had been included. The following text has been included in the manuscript:

Identifying this type of loss, as well as the associated emotions and barriers to processing it, is an essential step towards preventing this grief from becoming chronic. This information, together with the information on caregiver overload, could be of great utility in designing programmesto address the specific needs of this population. The specific objective was to analyse the following: the manifestations of grief in the relatives of individuals with ASD; their reactions to the diagnosis (associated thoughts and emotions); and how this has affected their lives (family, work, social, as well as relationships and personal changes).

Linked to this point, more details are required on the questions and prompts asked of parents during the interviews.  Did the same person conduct all the interviews?

The following text and a table have been included in response to the reviewer’s comments:

Subsequently, family members were interviewed. All of the interviews were conducted by the same trained researcher, specialized in the field of health psychology,on the mental health clinics and associations’ premises, in a quiet environment ensuring privacy.

The interviewer is an expert in qualitative methodology. In addition, a pilot study was conducted, which consisted of interviewing two professionals from the participating centres.

The one-to-one interviews were semi-structured and largely based on previous research in the area. Although the main aim of the interview was to explore patterns in going through the grieving process, it seemed sensitive to gather further data on other issues, such as information about the moment of receiving the diagnosis, the caregiver's overload or the change in future expectations. Table 2 shows the general framework of the four sections in which the interview was structured.

Table 2. Sections of the semi-structured interview.

Section 1: Diagnosis

Who communicated the diagnosis to the relative? At what age? How did you feel at the moment you were communicated? What were your reactions? What concerns came to mind?

Section 2: Loss and grieving process

Interference in work, leisure, friendship, family, care people and relationship, health, welfare in general

Associated emotions

Perceived losses

Section 3: Caregiver overload

Activities that you used to do and have stopped doing now.

Do you spend time for yourself? How much per week and how?

How has it affected family dynamics?

Section 4: Future

Has your vision changed about the future of your family member due to diagnosis? Could you explain it?

What prior contact did they have with the parents?

The following text has been added in the methods section:

Centre staffs were informed of the research objectives and they subsequently contacted theparticipants. The participants were called to a meeting where the principal investigator informed them about the study and answered their questions. Interested individuals signed the informed consent form.

What information was given to parents about the reasons for undertaking the study and how their information would be used? 

In response to this question, we would like to clarify the following:

In the information sheet, participants were provided with general data of the research, the voluntary nature of the participation, the commitment acquired when participating, possible risks and benefits for the participant, the confidential nature of the data, the rights of the participant and the data of contact of the main researcher.

The core idea of ​​this study is to contribute to the understanding of the grieving process that the relatives of a child with ASD go through, as well as to become aware of the emotions they experience, with the objective of being able to design intervention and bereavement programs for these relatives.

However, we have included the following paragraph in the text:

In the information sheet, participants were provided with general data of the research, the voluntary nature of the participation, the commitment acquired when participating, possible risks and benefits for the participant, the confidential nature of the data, the rights of the participant and the data of contact of the main researcher.

What additional support was available to parents if they found the interview process upsetting?  Did any parents become emotionally upset?

The issues addressed during the interview were emotionally important for family members. They found during the interview a space of liberation to talk about issues that can hardly be addressed in other contexts. Interviews served thus themselves as a therapeutic tool as they provided the opportunity to participants to release difficult emotions and thoughts not expressed until then.

We have included the following text:

The procedure included providing emotional support after the interview to the family members who needed it or requested it. However, in no case was it necessary to do so.

The sample of parents span a wide range of ages but no information is to given about the ages of children nor of their current circumstances, for example are they living with their parents?  

The following text has been included:

The age range of the children at the time of the study was from 6 to 12 years of age (mode = 8). All of the children were attending school and lived with their original families (15 with both of their parents and 5 with their mothers).

Although information is provided about the professional who gave the diagnosis, it is more crucial to know how many years ago the diagnosis was given. Although I appreciate that grief reactions can re-occur, I hope there were a majority of parents for whom diagnosis had occurred in the recent past.  Was this a requirement in the recruitment strategy?

An inclusion criterion set was for the period of time since the communication of the diagnosis to be at least one and a half years. However, no potential participants had to be excluded due to this criterion, and the shortest period of time since diagnosis communication of any of the participants was two years.

More detail is needed about the approach used in the qualitative data analysis, such as how grounded theory was undertaken?  What evidence did the authors find that grief emerged  as a core theme in parental responses?  Were there instances of   how parents coped with their grief?  Also what checks were used to assess agreements across raters and were the emerging themes confirmed with parents?

In order to answer these questions, the following text has been included:

Specifically, a method of constant comparison was used, organized into three stages, adapting the proposal of Strauss and Corbin(2002) to the present research objectives. First, an inductive strategy was used in which fragments of the participants’ discourse were analyzed and codified (micro-analysis). The properties of each code were later described and delimited, allowing for comparison among them. Names for these codes were selected based on expressions in the participants’ discourse. This step represented the open coding stage, in which four researchers participated. The different codes were named and defined by consensus. In cases of dis-agreement, debate among the researchers and a review of the literature were conducted to ensure that each code was as close as possible to the experience of the participants. Second, the identified codes (see Table 2) were grouped into categories according to their characteristics. These categories were derived from the participants’ discourses and were created by comparing cases and merging codes. The most complex categories were organized into subordinate concepts or sub-categories (axial coding). The same strategy as in the previousn analytic phase was adopted for grouping the codes. Finally, the categories obtained were integrated, and hierarchical relations were discussed. A central (core) category was identified and the most important codes were used to build a theoretical framework to explain the phenomenon.A core category must be related to other categories, must appear frequently in the data, and must provide a coherent explanation of the data (Strauss & Corbin, 2002). Later, the theory was refined using strategies of constant comparison. The final model was submitted to triangulation among all of the investigators to ensure its suitability for the phenomenon under study. This final stage corresponds to selective coding as described by Strauss and Corbin (2002).The interviews were transcribed by the researchers with wide experience in qualitative investigation. The analysis was performed using the original Spanish interviews. The quotations selected for inclusion in this article were independently translated by two experienced professional translators (English-native and Spanish-native), who then produced a jointly-agreed version that was further reviewed by two of the authors (both fluent in English) to ensure that the meaning of the original was accurately conveyed.

The discussion is largely a recap of the findings when it should be more focused on the specific aims of the study (see previous comment) and the implications for services and professional involved in conveying a diagnosis to parents in both the immediate term and also over time.  However they also need to discuss the value of utilising grief as an explanatory concept for parental responses when there are other frameworks available, especially ones that relate to present circumstances rather than events that may have taken place many years previously

In response to these recommendations, the following text has been removed:

In most cases, the diagnosis was communicated by a psychologist at an early age (44.89 months on average). An interesting result is that, in this study, some family members observed signs and/or signals that made them suspect and/or become alerted to the possibility that their family member was affected by a disorder. In these cases, when the professional informed them of the diagnosis, the relatives were already expecting it, and only feared the confirmation of their suspicions. Also, the initial impact of the diagnosis was lower, and in some cases we even found a proactive attitude towards it.

With regard to the disruption of expectations, the results were very homogeneous. Most family members reported that, during the first year of life, they had not suspected that their child might have an ASD-related problem, and if they did, they were unaware of everything involved in having a family member with this disorder. They had envisioned that their children would have had a typical development, and thus had normal expectations related to the different stages and aspects of their children’s lives.

The deficit in the communicative and social functioning distinctive of these children is a barrier that hinders and determines social interactions, as well as how this deficit is managed

In short, family members of individuals with ASD find themselves without support, affected by their losses, without the right to express it, and with high levels of overload, stress, sadness, and grief, manifestations which can end up causing physical and mental health disorders and illnesses.   

The following text has been included:

In summary, the following may be highlighted among the main contributions deriving from this study: a) the importance of making the diagnosis as soon as possible in order toallow family members to accept the disorder and cope with the new situation as soon as possible; b) the importance for these families of being able to identify some of the manifestations they experience as part of their grieving process, which is fundamental to overcoming it; c) it is important for professionals to be trained and proficient in identifying and recognising the emotional impact generated by communicating an ASD diagnosis and to be able to assess associated emotions and, if possible, to monitor these emotions; andd)it is necessary to address the needs that have been identified in the parents’ discourses. These needs are related both to the normalisation of the grieving process (which in most cases is not recognised as such) and to the provision of coping strategies regarding the management of the disorder and the levels of caregiver overload.

Under limitations, I suspect that if some of the comments noted above cannot be addressed, they should be added to this section.

In the limitations section we have added some of the suggestions made by the reviewer:

A further major limitation of the present work is its cross-sectionalnature. It is necessary to continue to make progress in identifying the specific characteristics of this type of grief by focusing on its cyclical nature. To this end, it would be important to carry out longitudinal studies with long-term monitoring. In addition, the sampling method used was not randomized. Rather,intentional sampling was used, thus limiting the application of the results to other settings. Another limitation of this study is the low number of male participants. Finally, differences in the severity of the diagnoses could not be taken into account in order to assess their effect on the grieving process. As a result, caution must be exercised when interpreting the results obtained and generalising them to other populations in different cultural settings.

In the conclusion section, the authors might provide more detail on what specific actions services need to undertake in support of families.  In many countries, their focus is largely on the person with autism.

We understand that the text that has been added previously also responds to this comment made by the reviewer.

As previously indicated, the following text has been included:

In summary, the following may be highlighted among the main contributions deriving from this study: a) the importance of making the diagnosis as soon as possible in order toallow family members to accept the disorder and cope with the new situation as soon as possible; b) the importance for these families of being able to identify some of the manifestations they experience as part of their grieving process, which is fundamental to overcoming it; c) it is important for professionals to be trained and proficient in identifying and recognising the emotional impact generated by communicating an ASD diagnosis and to be able to assess associated emotions and, if possible, to monitor these emotions; andd)it is necessary to address the needs that have been identified in the parents’ discourses. These needs are related both to the normalisation of the grieving process (which in most cases is not recognised as such) and to the provision of coping strategies regarding the management of the disorder and the levels of caregiver overload.

Reviewer 2 Report

General Comments:

I had the pleasure to review the manuscript (MS) entitled ‘Ambiguous grief experiences in family caregivers of children with autism spectrum disorder (ASD)‘. The manuscript, in general, is well written, with a solid theory supporting the research evidence, and requires careful examination. I believe, few more changes are required and few issues have to be examined from the research team, before the MS is ready for re submission and publication.

Let me now be more specific and present you my comments in detail.

Specific:

Title: The term ‘Ambiguous’ may not be present in the title (although ambiguous loss is the main terminology used in the theory of Boss, 1999). The term appears in the findings (see table in P10 for example, or P9 L206), as an outcome of the analysis conducted. ‘Grief Experiences’, ‘Feelings of Grief’, or even deleting the term ‘Ambiguous’ may be a more appropriate way for the title to appear (see P2 L61, 62: feelings of loss, feelings of grief, etc). Besides that, the term ‘ambiguous’ refers to both positive and negative feelings, which is not the case in the present study (some positive feelings are presented only at the end of the MS: e.g P11 L274).

Consider combining paragraphs 1 and 2

P1 L34: What does ‘numerous’ stands for? I do not follow the term ‘numerous syndromes’. Please, be specific.

P1 L39: What does ‘all of these needs’ mean? I assume it is the needs described above, but …… assuming is not good enough. It is the begin of the paragraph and the researchers have to declare carefully what they mean.

P2 L80-83: OK, here is what is missing from the MS: ‘… families did not experience the stages of bereavement only once; rather, they continually revisited the different stages again and again, especially when faced with developmental milestones that were not being reached by their children’. To my understanding, as much as I can follow the theory, the grief and loss experiences are going through a dynamic stage. That means they change across time, and certain variables may be involved into this dynamic process. This is the major weakness of the MS, it does not describe the changes across time, and does not describe the exact stage the caregivers experienced during the data collection process. I went through the literature, briefly, and I found some evidence in the web. I am attaching this information for the researchers to consider. To my understanding, it is essential for the researchers to attempt to recognize the exact stage of the caregivers for the purposes of the study. Strategies and planning for future support may not overcome this obstacle. For that purpose, the researchers may also consider the work of Marinelli and Dell Orto (1999).

P3 L100: OK, I see some demographics here and variables that may have an impact upon the families and the grief experiences they are going through. Need also to have an idea upon the severity of the ASD diagnosis. The last (severity) is very important, indeed. The findings may be strongly associated with the demographic variables of the caregivers (e.g. income).

Methods: Here is some information that must be incorporated into the MS. The goal of my criticism is only to assist the researchers to improve the quality of their work. Therefore, I need to ask certain questions and expect the responses to be provided in the revised manuscript.

Who conducted the interviews? Was this person(s) qualified enough to collect qualitative data? What was his/her (their) previous training? Did he/she run a pilot study first? The ‘Author Contributions’ (P15 L424), at the end of the MS, does not incorporate this information. Further, triangulation of the data analysis is missing. Some type of credibility and trustworthiness is missing too. We need to have an idea with respect to the methods followed to analyze data (similarly to validity and reliability evidence). Otherwise the MS findings are questionable and subjected to criticism.

Further, the questionnaire used is essential to consider. We need to have some information with respect to the preparation of the questions, the theory attached, the pilot testing, the responsible team for preparing the research tool, etc.

P5 L144: ‘Some’ is not useful, please be specific.

P6 L174: ‘severity’ (of ASD diagnosis) is referred to and may be considered as a variable of paramount importance.

P9 L206: The term ‘ambiguous loss’ appears. Well, I do not know if this is good enough for the term to be incorporated it into the title (reappears under ‘Gains’ P11 L274, 280, 285).

P12 L289: ‘We found’…. Who did? Rephrase please.

Finally, the limitations section must be improved, to a wide extent. The sample size is not the only limitation. Actually, in a qualitative study, with semi-structured interviews, the examination of 20 participants is not at all a small figure. This part of the MS was prepared in a hurry and must be improved.

Web Information:

Marinelli R, & Dell Orto A. (1999). The Psychological and Social Impact of Illness and Disability. New York, NY: Springer Publishing Company.

https://www.canr.msu.edu/news/ambiguous_loss

……..the five stages of grief: Denial, anger, bargaining, depression and acceptance. Since then there have been numerous books and articles written on the subject of grief and loss

Author Response

We are very grateful to the reviewers for their careful reading of our manuscript Ambiguous grief experiences in family caregivers of children with autism spectrum disorder (ASD).

We include below our point-by-point responses to the reviewers’ comments with an account of the corresponding modifications.

Thank you for your attention.

Specific:

Title: The term ‘Ambiguous’ may not be present in the title (although ambiguous loss is the main terminology used in the theory of Boss, 1999). The term appears in the findings (see table in P10 for example, or P9 L206), as an outcome of the analysis conducted. ‘Grief Experiences’, ‘Feelings of Grief’, or even deleting the term ‘Ambiguous’ may be a more appropriate way for the title to appear (see P2 L61, 62: feelings of loss, feelings of grief, etc). Besides that, the term ‘ambiguous’ refers to both positive and negative feelings, which is not the case in the present study (some positive feelings are presented only at the end of the MS: e.g P11 L274).

We have made the following change to the title:

Grief experiences in family caregivers of children with autism spectrum disorder (ASD).

Consider combining paragraphs 1 and 2

We have made the following change to the text:

An estimated 1 in 160 children has an autism spectrum disorder (ASD). This is the estimated mean, as the observed prevalence varies considerably across studies, with well-controlled studies reporting considerably higher figures [1]. According to epidemiological studies conducted over the past 50 years, the global prevalence of these disorders seems to be increasing. Some possible explanations for this apparent increase have been suggested, including greater awareness, broader diagnostic criteria, better diagnostic tools, and improved communication among professionals [1].

P1 L34: What does ‘numerous’ stands for? I do not follow the term ‘numerous syndromes’. Please, be specific.

We agree with your clarification and have modified that sentence.The following has been included:

ASDs is a range of mental disorders of the neurodevelopmental type that appear at an early age and are characterised by difficulties in social interaction, communication, and flexibility, among others [2].

P1 L39: What does ‘all of these needs’ mean? I assume it is the needs described above, but… assuming is not good enough. It is the begin of the paragraph and the researchers have to declare carefully what they mean.

This sentence has been modified to read as follows:         

Planning for all of these needs starts with one of the key moments: informing the family of the diagnosis.

Planning the needs discussed above starts with one of the key moments: informing the family of the diagnosis.

P2 L80-83: OK, here is what is missing from the MS: ‘… families did not experience the stages of bereavement only once; rather, they continually revisited the different stages again and again, especially when faced with developmental milestones that were not being reached by their children’. To my understanding, as much as I can follow the theory, the grief and loss experiences are going through a dynamic stage. That means they change across time, and certain variables may be involved into this dynamic process. This is the major weakness of the MS, it does not describe the changes across time, and does not describe the exact stage the caregivers experienced during the data collection process. I went through the literature, briefly, and I found some evidence in the web. I am attaching this information for the researchers to consider. To my understanding, it is essential for the researchers to attempt to recognize the exact stage of the caregivers for the purposes of the study. Strategies and planning for future support may not overcome this obstacle. For that purpose, the researchers may also consider the work of Marinelli and Dell Orto (1999).

Regarding the reviewer’s comments on the stages of bereavement, the family members’ revisiting of the stages of bereavement in the face of their children’s developmental milestones is mentioned in the introduction to explain the concept of ambiguous loss within the literature review. We agree with the reviewer that grief is a dynamic process. While it is true that the description of grief in stages is useful for assisting people after a loss experience, the framework for our work is William Worden’s tasks model (2002). From this point of view, what is important is to identify which of the tasks the family members have not been able to perform in order to process their grief. We are referring to the following: a) the acceptance of the reality of their loss (which is difficult in disallowed grief); b) the identification and expression of the emotions associated with their loss (which constitutes a fundamental part of the interviews in this article); c) adaptation to the new reality, measured through interference; and d) being able to set new goals and experience more leisure time. It is therefore not the aim of our work to determine which stage of bereavement each of the participants is in. We are grateful for the reference works you suggested to us.

P3 L100: OK, I see some demographics here and variables that may have an impact upon the families and the grief experiences they are going through. Need also to have an idea upon the severity of the ASD diagnosis. The last (severity) is very important, indeed. The findings may be strongly associated with the demographic variables of the caregivers (e.g. income). 

We do agree that there are many variables that can influence the grieving process, as well as the acceptance of the diagnosis. However, it is not clear from the participants’ discourses that there are differences according to the caregivers’ demographic variables (socio-economic level: 80% of the families were middle-income families), nor according to the severity of the diagnosis. However, we have included this particular consideration in the limitations of our work. Methods: Here is some information that must be incorporated into the MS. The goal of my criticism is only to assist the researchers to improve the quality of their work. Therefore, I need to ask certain questions and expect the responses to be provided in the revised manuscript.

Who conducted the interviews? Was this person(s) qualified enough to collect qualitative data? What was his/her (their) previous training? Did he/she run a pilot study first?

The following text has been included to answer these questions:

Subsequently, family members were interviewed. All of the interviews were conducted by the same trained researcher, specialized in the field of health psychology, on the mental health clinics and associations’ premises, in a quiet environment ensuring privacy.

The interviewer is an expert in qualitative methodology. In addition, a pilot study was conducted interviewing two professionals from the participating centers.

The one-to-one interviews were semi-structured and largely based on previous research in the area. Although the main aim of the interview was to explore patterns in going through the grieving process, it seemed sensitive to gather further data on other issues, such as information about the moment of receiving the diagnosis, the caregiver's overload or the change in future expectations. Table 2 shows the general framework of the four sections in which the interview was structured.

The ‘Author Contributions’ (P15 L424), at the end of the MS, does not incorporate this information.

We had not provided information on this issue because this parameter is not included among the items regarding the author contributions listed by the journal. The first author, J.B., conducted the interview.

Further, triangulation of the data analysis is missing. Some type of credibility and trustworthiness is missing too. We need to have an idea with respect to the methods followed to analyze data (similarly to validity and reliability evidence). Otherwise the MS findings are questionable and subjected to criticism.

The following text has been included to answer these questions:

Specifically, a method of constant comparison was used, organized into three stages, adapting the proposal of Strauss and Corbin(2002) to the present research objectives. First, an inductive strategy was used in which fragments of the participants’ discourse were analyzed and codified (micro-analysis). The properties of each code were later described and delimited, allowing for comparison among them. Names for these codes were selected based on expressions in the participants’ discourse. This step represented the open coding stage, in which four researchers participated. The different codes were named and defined by consensus. In cases of dis-agreement, debate among the researchers and a review of the literature were conducted to ensure that each code was as close as possible to the experience of the participants. Second, the identified codes (see Table 2) were grouped into categories according to their characteristics. These categories were derived from the participants’ discourses and were created by comparing cases and merging codes. The most complex categories were organized into subordinate concepts or sub-categories (axial coding). The same strategy as in the previousn analytic phase was adopted for grouping the codes. Finally, the categories obtained were integrated, and hierarchical relations were discussed. A central (core) category was identified and the most important codes were used to build a theoretical framework to explain the phenomenon.A core category must be related to other categories, must appear frequently in the data, and must provide a coherent explanation of the data (Strauss & Corbin, 2002). Later, the theory was refined using strategies of constant comparison. The final model was submitted to triangulation among all of the investigators to ensure its suitability for the phenomenon under study. This final stage corresponds to selective coding as described by Strauss and Corbin (2002).The interviews were transcribed by the researchers with wide experience in qualitative investigation. The analysis was performed using the original Spanish interviews. The quotations selected for inclusion in this article were independently translated by two experienced professional translators (English-native and Spanish-native), who then produced a jointly-agreed version that was further reviewed by two of the authors (both fluent in English) to ensure that the meaning of the original was accurately conveyed.

Further, the questionnaire used is essential to consider. We need to have some information with respect to the preparation of the questions, the theory attached, the pilot testing, the responsible team for preparing the research tool, etc.

The following text and a table has been included to answer these questions:

The one-to-one interviews were semi-structured and largely based on previous research in the area. Although the main aim of the interview was to explore patterns in going through the grieving process, it seemed sensitive to gather further data on other issues, such as information about the moment of receiving the diagnosis, the caregiver's overload or the change in future expectations. Table 2 shows the general framework of the four sections in which the interview was structured.

Table 2. Sections of the semi-structured interview.

Section 1: Diagnosis

Who communicated the diagnosis to the relative? At what age? How did you feel at the moment you were communicated? What were your reactions? What concerns came to mind?

Section 2: Loss and grieving process

Interference in work, leisure, friendship, family, care people and relationship, health, welfare in general

Associated emotions

Perceived losses

Section 3: Caregiver overload

Activities that you used to do and have stopped doing now.

Do you spend time for yourself? How much per week and how?

How has it affected family dynamics?

Section 4: Future

Has your vision changed about the future of your family member due to diagnosis? Could you explain it?

In addition to the design of the interview, a previous study on the grieving of parents with children with cerebral palsy served as the basis:

Fernández-Alcántara, M., García-Caro, M. P., Laynez-Rubio, C., Pérez-Marfil, M. N., Martí-García, C., Benítez-Feliponi, Á., ... & Cruz-Quintana, F. (2015). Feelings of loss in parents of children with infantile cerebral palsy. Disability and health journal8(1), 93-101. 

P5 L144: ‘Some’ is not useful, please be specific.

The following sentence has been replaced:

Some family members had already been observing certain signs that made them suspect and/or become alerted to the possibility that the child might be affected by some disorder.

by the following sentence:

Most of the mothers had already been observing certain signs that made them suspect and/or become alerted to the possibility that the child might be affected by some disorder.

P6 L174: ‘severity’ (of ASD diagnosis) is referred to and may be considered as a variable of paramount importance

We have already replied to this question regarding the comments P3 L100.

P9 L206: The term ‘ambiguous loss’ appears. Well, I do not know if this is good enough for the term to be incorporated it into the title (reappears under ‘Gains’ P11 L274, 280, 285).

We have changed the title.

P12 L289: ‘We found’…. Who did? Rephrase please.

We have replaced the following sentence:

We found that there is a time sequence that begins with the anticipation of the diagnosis, before receiving it officially.

with the following sentence:

The results show that there is a time sequence that begins with the anticipation of the diagnosis, before receiving it officially.

Finally, the limitations section must be improved, to a wide extent. The sample size is not the only limitation. Actually, in a qualitative study, with semi-structured interviews, the examination of 20 participants is not at all a small figure. This part of the MS was prepared in a hurry and must be improved.

In the limitations section we have added some of the suggestions made by the reviewer:

A further major limitation of the present work is its cross-sectionalnature. It is necessary to continue to make progress in identifying the specific characteristics of this type of grief by focusing on its cyclical nature. To this end, it would be important to carry out longitudinal studies with long-term monitoring. In addition, the sampling method used was not randomized. Rather,intentional sampling was used, thus limiting the application of the results to other settings. Another limitation of this study is the low number of male participants. Finally, differences in the severity of the diagnoses could not be taken into account in order to assess their effect on the grieving process. As a result, caution must be exercised when interpreting the results obtained and generalising them to other populations in different cultural settings.

Reviewer 3 Report

Introduction

The article is very well written. The question is relevant, current and well founded theoretically and empirically. The authors show that they have an exhaustive knowledge of the subject. The documentation is abundant but adequate to justify the objective.

Methodology.

The number of participants is enough and normally with a smaller number of data the saturation is usually reached. The reasons for the difference between men and women are not mentioned in this section, although this limitation (few men are mentioned as one of the limitations of the study in the discussion section). The procedure is sufficiently operationalized as well as the analysis of data. In line 122, the expression "...the scientific literature, specifically..." can be deleted. It would look like this: "A qualitative design was used based on the grounded..."

Results

The results are sorted. The themes that emerge are exclusive, well defined and the examples chosen are exemplary. As for the arithmetic means that appear in lines 99 and 162, it would be more appropriate to use the median given the low number of cases and the excessive influence that extreme cases can have, such as the age in months of cases 9 and 10 in table 3.

In the results section, the synthesis presented in section 3.1.1 stands out, as well as the conceptual structure provided by figure 1.

Discussion.

On line 328 change the value of the arithmetic mean to the median. The results are duly discussed under the protection of existing evidence on the subject and the theoretical framework. Again, the authors show their great knowledge of the subject which translates into a very adequate discussion of the results. In line 408 the authors mention that the main limitation of his study is the reduced number of participants in the sample. In my opinion, the sample is enough in reference to women because normally in qualitative research, with the methodology used, the information tends to reach theoretical saturation with a lower number of participants. In addition, the data in table 1 define the theoretical sampling that the authors have followed in the selection of participants in terms of age, educational level, marital status and employment status. If it constitutes a limitation, as the authors point out, the low number of men.

Author Response

We are very grateful to the reviewers for their careful reading of our manuscript Ambiguous grief experiences in family caregivers of children with autism spectrum disorder (ASD).

We include below our point-by-point responses to the reviewers’ comments with an account of the corresponding modifications.

Thank you for your attention.

Methodology.

The number of participants is enough and normally with a smaller number of data the saturation is usually reached. The reasons for the difference between men and women are not mentioned in this section, although this limitation (few men are mentioned as one of the limitations of the study in the discussion section). The procedure is sufficiently operationalized as well as the analysis of data.

We are grateful for the reviewer’s comments.

In line 122, the expression "...the scientific literature, specifically..." can be deleted. It would look like this: "A qualitative design was used based on the grounded..."

We have included the reviewer's comment in the text:

A qualitative design was used based on the scientific literature, specifically on the grounded theory by Glaser & Strauss (1967).

Results

The results are sorted. The themes that emerge are exclusive, well defined and the examples chosen are exemplary. As for the arithmetic means that appear in lines 99 and 162, it would be more appropriate to use the median given the low number of cases and the excessive influence that extreme cases can have, such as the age in months of cases 9 and 10 in table 3.

The text has been modified. The median has been included and the mean has been removed.

The median age of these individuals was 40.00 years (range: 31-68).

In the present study, the children’s median age at which the parents were informed of the diagnosis of ASD was 30.00 months (range: 17-132).

In the results section, the synthesis presented in section 3.1.1 stands out, as well as the conceptual structure provided by figure 1.

Thank you very much for your comments.

Discussion.

On line 328 change the value of the arithmetic mean to the median.

The text has been modified. The median has been included and the mean has been removed.

The results are duly discussed under the protection of existing evidence on the subject and the theoretical framework. Again, the authors show their great knowledge of the subject which translates into a very adequate discussion of the results.

In line 408 the authors mention that the main limitation of his study is the reduced number of participants in the sample. In my opinion, the sample is enough in reference to women because normally in qualitative research, with the methodology used, the information tends to reach theoretical saturation with a lower number of participants.

In addition, the data in table 1 define the theoretical sampling that the authors have followed in the selection of participants in terms of age, educational level, marital status and employment status. If it constitutes a limitation, as the authors point out, the low number of men.

In the limitations section we have added some of the suggestions made by the reviewer:

A further major limitation of the present work is its cross-sectionalnature. It is necessary to continue to make progress in identifying the specific characteristics of this type of grief by focusing on its cyclical nature. To this end, it would be important to carry out longitudinal studies with long-term monitoring. In addition, the sampling method used was not randomized. Rather,intentional sampling was used, thus limiting the application of the results to other settings. Another limitation of this study is the low number of male participants. Finally, differences in the severity of the diagnoses could not be taken into account in order to assess their effect on the grieving process. As a result, caution must be exercised when interpreting the results obtained and generalising them to other populations in different cultural settings.

Round 2

Reviewer 1 Report

I appreciate the revisions which the authors have undertaken and the paper is all the stronger for them.  

One small revision.  Details of the age of diagnosis and the professionals who made it should be moved to the participant section.  In the the findings section, reference can be made to the table.

Author Response

We are very grateful to the review for your careful reading of our manuscript grieving experiences in family caregivers of children with autism spectrum disorder (ASD).

We have included suggested revisions in the text.

Thanks for your attention.
